# Reversible Bronchial Obstruction in Primary Ciliary Dyskinesia

**DOI:** 10.3390/jcm11226791

**Published:** 2022-11-16

**Authors:** Hagit Levine, Ophir Bar-On, Vered Nir, Nicole West, Yotam Dizitzer, Huda Mussaffi, Dario Prais

**Affiliations:** 1Pulmonary Institute, Schneider Children’s Medical Center, Petah-Tikva 49100, Israel; 2Sackler Faculty of Medicine, Tel Aviv University, Tel Aviv 6997801, Israel; 3Department of Pediatrics, Hillel-Yaffe Medical Center, Hadera 3810101, Israel; 4Department of Pediatrics, Schneider Children’s Medical Center, Petah-Tikva 4920235, Israel

**Keywords:** primary ciliary dyskinesia, airway hyperreactivity, airway obstruction, asthma, bronchodilators, inhaled corticosteroids, spirometry

## Abstract

Background: Inhaled bronchodilators are frequently used among patients with primary ciliary dyskinesia (PCD), although neither the effectiveness nor the prevalence of their use is known, due to the paucity of relevant studies. Methods: This is a retrospective analysis of pre- and post-bronchodilator spirometry results, of patients with PCD from two centers. Correlations were examined of bronchodilator response, with asthma and atopy markers. Results: Of 115 patients, 46 (40%) completed spirometry pre- and post-bronchodilation. Of these, 26 (56.5%) demonstrated reversible airway obstruction (increase in %FEV_1_ predicted ≥ 10%). Obstruction reversibility was not found to be associated with a family history of asthma, blood eosinophil level, elevated IgE, or atopy symptoms. Of the 46 patients who completed bronchodilator spirometry, 29 (63%) were regularly using bronchodilators and inhaled corticosteroids. Conclusions: More than half of patients with PCD presented with reversible airway obstruction, without any correlation to markers of personal or familial atopy. Inhaled bronchodilators and corticosteroid therapies are commonly used for treating PCD. Evaluating bronchodilator response should be considered, and its effectiveness should be further studied.

## 1. Introduction

Primary ciliary dyskinesia (PCD) is an uncommon, genetically heterogeneous disorder, with a prevalence of approximately 1 in 15,000 births. The inheritance is usually autosomal-recessive and results in dysfunction of motile cilia, leading to mucus stasis. Mutations in more than 40 genes have been reported to cause PCD; the involvement of many other genes is likely to be discovered. As PCD is more common in populations with closed genetic pools, genetic heterogeneity is seen in socially isolated consanguineous populations. Estimates of prevalence are scarce in populations outside of Europe, but the disease is expected to be more common in certain populations, such as in Arab countries [1].

In the lungs, PCD manifests as a chronic progressive airway disease starting in the neonatal period or during infancy, which then gradually advances to suppurative lung disease with bronchiectasis in adult life [1,2]. About 50% of patients have situs inversus (Kartagener’s syndrome). Situs ambiguus, including heterotaxy, is reported in up to 12% of patients. Chronic rhinosinusitis, recurrent otitis media until conductive hearing impairment and infertility are also common [1]. Airway pathology stems from a dysregulated cycle of infection (with pathogenic bacteria including *Pseudomonas aeruginosa*) and inflammation [3]. Indeed, airway inflammation is common in PCD, as in cystic fibrosis (CF) and asthma; however, the pathomechanisms are different. In PCD, the inflammation is mostly neutrophilic; whereas in asthma, eosinophilic Th2 inflammation is more common [4].

Individuals with PCD frequently use inhaled beta2-agonists, with or without saline inhalations, to soothe shortness of breath and to enhance bronchodilation before chest physiotherapy. However, the effectiveness of bronchodilators in PCD has not been documented, due to the paucity of relevant studies [5].

Our study aimed to identify the frequency of airway reversibility following inhaled bronchodilators in individuals with PCD. Furthermore, we aimed to correlate bronchodilator response with markers of personal or family atopy.

## 2. Methods

This study reviewed medical charts of patients diagnosed with PCD who were followed at Schneider Children’s Medical Center and Hillel Yaffe Medical Center. Inclusion criteria: a diagnosis of PCD, based on electron microscopic examination or genetic diagnosis of a defect in the dynein arm, according to accepted criteria; [2] and spirometry performed before and after inhaled bronchodilators, according to ATS/ERS guidelines [6,7]. 


Standardization of pulmonary function tests:


Spirometry, performed both before and after inhaled bronchodilators, was evaluated during routine follow-up visits. Spirometry was also conducted during exacerbations that were marked by symptoms such as wheezing or dyspnea. For the purpose of this study, the test that demonstrated the largest increase in %FEV_1_ following bronchodilator administration was used. 

Prediction equations that were used by both centers were the Knudson or ECCS/ERS for patients aged 18 years and above, and Polgar for children younger than age 18 years. All the patients followed standard instructions before performing spirometry, namely avoiding the use of bronchodilators (short-acting beta2 agonists (SABA) at least 6 h prior the test, and long-acting beta-agonists (LABA)/combined LABA and inhaled corticosteroids (ICS) at least 12 h prior the test). 


Definition:


Reversible airway obstruction was defined as an increase in the percentile predicted FEV_1,_ (%FEV_1_) by 10% or more [7]. 


Patient data:


The data collected included demographic details and measurements of weight and height at the best-recorded increase in %FEV_1_ on spirometry. The atopy markers considered were: known allergies, the highest value of serum IgE, and eosinophil count. First-degree relative history of asthma was also documented. Other medical information collected from the patient files included: respiratory symptoms after birth, the presence of situs-inversus, recurrent wheezing, and the presence of consanguinity. Lastly, we documented whether patients were receiving inhaled bronchodilators: SABA, LABA or ICS. 


Statistics:


Demographic and clinical data were described using medians and ranges, or means and standard deviations, as appropriate, according to group characteristics. To identify differences between groups, Pearson’s chi-square test or Fisher exact test (two-tailed) were used for categorical variables, and *t*-tests for normally distributed continuous variables. The Mann–Whitney test was used for the distribution of continuous variables (such as age across categories). Significant differences were defined as a *p*-value (a two-sided alpha level) ≤ 0.05. The statistical analysis was performed with SPSS 22.0 (IBM, Armonk, NY, USA) for Windows. 

The Rabin Medical Center and Hillel-Yaffe Medical Center Research Ethics Committees approved this retrospective study (*Rabin IRB number:* 0392-17-RMC; *Hillel-Yaffe IRB number*: 0087-22-HYMC).

## 3. Results

Of 115 patients with PCD who were followed during the study period, 46 (40%) fulfilled the inclusion criterion of at least one recorded spirometry test pre- and post-bronchodilator use. Demographic and anthropometric characteristics are presented in Table 1. The patients who were excluded were too young to perform satisfactory spirometry or had not performed spirometry following bronchodilation. The median age at the best-recorded spirometry was 12 years; the range was 5–48 years.


Airway obstruction reversibility:


Airway obstruction reversibility following inhaled bronchodilators, expressed as an increase larger than 10% in %FEV_1_ of that predicted, was documented in 26 (56.5%) patients. The median %FEV_1_ at baseline, predicted for age, was 72% (range 32–109%) among all the patients, and 60% among those with bronchodilator reversibility (range 32–92%) during the best-recorded spirometry. The median change pre- and post-bronchodilator use was 16% (range 10–27%). Significant differences were not observed in the proportions of patients with airway reversibility, between males and females, between ethnic groups (Jewish vs. non-Jewish), nor between patients with and without documented familial consanguinity (Table 1).

The median age was 13.5 (5–41) years among the patients with %FEV_1_ predicted ≥ 10%, and age 11 (5–48) years among those with %FEV_1_ predicted < 10%; this difference was not statistically significant. In an analysis that matched patients according to age, airway reversibility correlated with worse %FEV_1_ predicted (*p* = 0.039) (Figure 1). 


Asthma and atopy correlation:


Prevalences of self-recorded asthma and of asthma among first-degree family members did not differ between patients with and without reversible airway obstruction (Table 2). Family asthma data were missing for some patients, yet laboratory parameters of atopy were available for most. Nonetheless, differences were not detected in maximal IgE levels, or in eosinophil counts, between patients with and without reversible bronchial obstruction. 

Significant associations were not found of prior transient respiratory symptoms in the newborn or of neonatal pneumonia, with airway reversibility. Notably, however, 91.7% of the patients with documented airway reversibility also had previous documentation of recurrent wheezing (*p* = 0.010).


The association of reversible airway obstruction with medication use


Of the 46 patients evaluated, 29 (63%) were using bronchodilators (Figure 2). Of them, 21 (72%) demonstrated reversible airway obstruction. Of the 26 patients with reversible airway obstruction, 21 (80%) were using inhaled bronchodilators (*p* = 0.005) (Figure 2). 

ICS usage was also common among the patients. Of the 46 with available data, 26 (56.5%) were using ICS therapy regularly. Of them, 19 (73%) demonstrated reversible airway obstruction. Of the 26 patients with reversible airway obstruction, only 7 (27%) were not using ICS therapy (*p* = 0.01) (Figure 2). 

## 4. Discussion

In this study of individuals with PCD, post-bronchodilator reversible airway obstruction was a common finding. Reversibility of airflow was found to be significantly associated with prior documented recurrent wheezing. However, correlations were not found of reversible bronchial obstruction with atopy, elevated IgE, increased eosinophils, or a family history of atopy. This study highlights the relations of the consistent use of bronchodilators and ICS therapy, with the presence of reversible airway obstruction. 

Of note, Phillips et al. [8] did not find a significant difference in the airway response to exercise after bronchodilators (Salbutamol), between healthy people and people with PCD. However, only 12 people were included in each group.

Airway reversibility was previously reported in about half of individuals with CF who were receiving bronchodilator therapy [9]. Galodé et al., reported higher prevalence of airway reversibility among younger patients: 73.5%, 48.5%, and 52.9%, in the 6–8 year, 10–12 year, and 15–17 year age groups, respectively. [10] Mitchell et al., reported a positive response to methacholine in 51% of patients with CF, compared to 98% of patients with a single diagnosis of asthma. [11] In that study, different dose–response curves highlighted differences between the pathophysiologic mechanisms of the two diseases. Van Haren et al. suggested a possible mechanism of increased vagal bronchomotor tone in CF, after demonstrating a bronchodilator response to ipratropium bromide following exercise. [12] 

In contrast to the documentations in individuals with CF, [9] this study revealed no correlation between gender and airway reversibility among patients with PCD. However, similar to the study on CF, [9] this study found no correlations of airway reversibility with a family or personal history of asthma, or with atopy. The mechanisms suggested in both CF and PCD are not related to asthma. 

Bronchial airway reversibility has also been described in non-CF bronchiectasis. Singh et al. showed bronchodilator reversibility in 30.4% of patients with non-CF bronchiectasis, even after excluding those with asthma, allergic bronchopulmonary aspergillosis, and chronic obstructive pulmonary disease, to avoid false-positive cases [13]. Furthermore, Guan et al. observed that patients with other reasons for bronchiectasis, who exhibited significant bronchodilator reversibility, commonly shared a few important characteristics: a higher bronchiectasis severity index, a higher prevalence of *Pseudomonas aeruginosa* isolation and infection, and poorer lung function at baseline [14]. Therefore, chronic infection has been suggested as responsible for bronchial hyperreactivity in bronchiectasis [15]. Given its clinical significance, Bulcun et al., proposed that bronchial hyperreactivity should be considered part of routine clinical evaluation in all patients with bronchiectasis [16].

The reason and mechanism for airway reversibility after bronchodilator use in bronchiectasis, and in PCD specifically, is still unclear. The mechanism responsible for this reversibility is probably different from that in patients with asthma or CF. One hypothesis relates to chronic airway inflammation, and effects of toxins, through infected or inflamed bronchial mucosa, on bronchial muscle cells. Bronchial hyperreactivity possibly affects the clearance mechanism, and perpetuation of the cycle of colonization of microbes and subsequent inflammation [17].

Another hypothesis suggests that bronchodilators are responsible for slightly improving the residual function of cilia in patients with mild cilia mutations and symptoms. Bronchodilators are known for their ability to enhance mucociliary clearance, [18] and could be beneficial for patients with PCD who have some residual ciliary movement. One might ask whether the mucus removal effect that clears the airways between spirometries, before and after inhaled bronchodilators, may be the cause for such difference. These hypotheses should be re-evaluated in vitro.

As no curative option for PCD is currently available, treatment is directed at preventing and managing disease complications. No recommendations have been issued for the administration of bronchodilators, ICS, or combination therapy in the guidelines of PCD treatment. [18] Some studies presented substantial heterogeneity in the management of PCD, within and between countries [19]. 

We report that over half our patients used bronchodilators, and a similar proportion used ICS therapy. Likewise, another study found that ICS are commonly prescribed in PCD, often without evidence of type 2 airway inflammation [20]. The benefits of long-term bronchodilators or ICS treatment in patients with PCD have not been demonstrated. 

Notably, the BESTCILIA multicenter trial showed dramatically reduced pulmonary exacerbations in patients with PCD treated with azithromycin than in a placebo group [21]. Macrolides are known to have bacteriostatic properties, as well as anti-inflammatory and immunomodulatory effects. Several studies have researched the use of azithromycin for treating other chronic respiratory suppurative diseases that are dominated by neutrophilic inflammation [22]. As the effectiveness of other treatments has recently been investigated, the long-term effects of bronchodilators and ICS should be further studied.

A strength of this study is its focus on objective measurements of maximal airway reversibility over the entire course of a patient’s documented history. Additionally, the patient data were accessed from two large PCD centers. 

Limitations of our study include those of any retrospective study design, the lack of a challenge test, variations in data availability among the numerous parameters investigated, and the limited utilization of pre- and post-bronchodilator spirometry. In addition, this study did not discern whether the maximal bronchodilator response occurred during a stable disease period, during an acute pulmonary exacerbation, or during a viral infection when a patient may be more prone to a positive response.

In addition to its presentation of the prevalence of bronchial reactivity among rare diseases such as PCD, this study raises the question as to whether spirometry tests before and after bronchodilators may lead to misdiagnosis of asthma instead of PCD. This issue is important due to the high frequency of spirometry performed worldwide, before and after bronchodilator use, to establish a diagnosis of asthma.

To conclude, reversible airway obstruction appears to be a central feature of PCD lung disease. At present, the associated pathophysiology is poorly understood. However, further research into this topic may inform future decision-making regarding inhaled bronchodilators and ICS therapy. Testing for bronchodilator response in bronchiectasis is important, as it can assess prognosis and guide treatment [14]. The presence of bronchodilator reversibility should be considered an indication for prescribing bronchodilators and ICS to all patients with PCD.

## Figures and Tables

**Figure 1 jcm-11-06791-f001:**
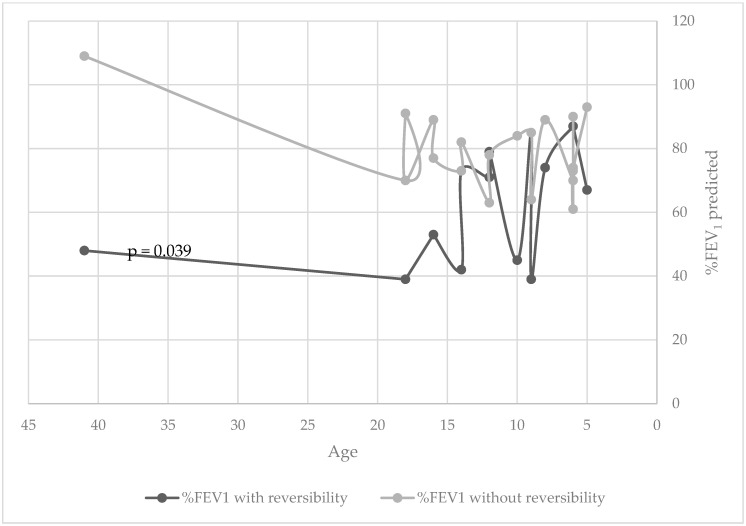
%FEV_1_ predicted for age comparing patient with primary ciliary dyskinesia with or without airway reversibility.

**Figure 2 jcm-11-06791-f002:**
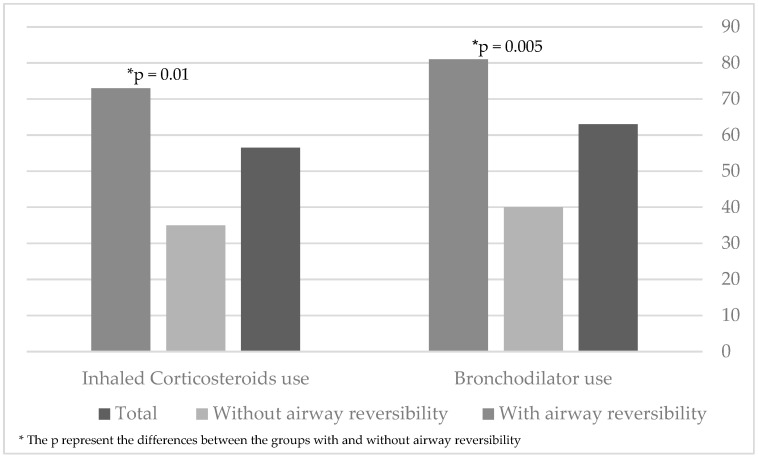
Inhaled bronchodilator and corticosteroid usage among patients with primary ciliary dyskinesia.

**Table 1 jcm-11-06791-t001:** Demographic and anthropometric characteristics of patients with primary ciliary dyskinesia, according to airway reversibility.

	Total *n* = 46 (100%)	Without Airway Reversibility *n* = 20 (43.5%)	With Airway Reversibility *n* = 26 (56.5%)	*p*-Value
**Sex (female)**	20 (43.5%)	7 (35.0%)	13 (50.0%)	0.309
**Ethnicity (*n* = 29/46, 63%)**				
**Jewish**	13 (44.8%)	7 (58.3%)	6 (35.3%)	
**Arabic**	15 (51.7%)	5 (41.7%)	10 (58.8%)	0.379
**Other**	1 (3.4%)	0	1 (5.9%)	
**Consanguineous (*n* = 28/46, 61%)**	14 (50.0%)	6 (50.0%)	8 (50.0%)	1.000
**Height (cm) (mean (SD))**	149 (19.6)	146.7 (22.0)	151.5 (16.3)	0.24
**Weight (kg) (mean (SD))**	44.6 (18.3)	42.9 (20.2)	46.6 (16.0)	0.345
**BMI (kg/m^2^) (mean (SD))**	19.1 (3.9)	18.6 (3.9)	19.7 (3.8)	0.899

**Table 2 jcm-11-06791-t002:** Respiratory and asthma/allergy characteristics of patients with primary ciliary dyskinesia, according to airway reversibility.

	Total *n* = 46 (100%)	Without Airway Reversibility *n* = 20 (43.5%)	With Airway Reversibility *n* = 26 (56.5%)	*p*-Value
**Spirometry:**				
**%FEV_1_ predicted** **(median, range)**	72 (32–109)	77 (61–109)	60 (32–92)	0.07
**Reversibility in %FEV_1_** **(median, range)**	11.5 (0–27)	5 (0–8)	16 (10–27)	<0.001
**Age (years) ** **(median, range)**	12 (5–48)	At best-recorded spirometry 11 (5–48)	At best reversibility 13.5 (5–41)	0.301
**Asthma/Atopy:**				
**Familial asthma**	6 (13.0%)	2 (10.0%)	4 (15.4%)	0.591
**Atopy/allergy**	4 (8.7%)	2 (10.0%)	2 (7.7%)	0.783
**Eosinophils ≥ 5%**	28 (62.2%)	14 (70.0%)	14 (56.0%)	0.336
**IgE ≥ 100 IU**	9 (20.0%)	5 (25.0%)	4 (16.0%)	0.453
** Neonatal/Childhood characteristics: **				
**Neonatal tachypnea or pneumonia (*n* = 25/46, 54%)**	19 (76.0%)	11 (91.7%)	8 (61.5%)	0.078
**Recurrent wheezing (*n* = 22/46, 48%)**	15 (68.2%)	**4 (40.0%)**	**11 (91.7%)**	**0.010**

ICS–inhaled corticosteroids.

## Data Availability

Not applicable.

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
