# Peer review of "Reversible Bronchial Obstruction in Primary Ciliary Dyskinesia"

_jcm, 2022, doi:10.3390/jcm11226791_

Round 1
Reviewer 1 Report
PCD is a rare and difficult for diagnosis condition. It comprises variety of phenotypes and multiple genetic variants. That is why every study, providing additional information on the topic is valuable. I think that the authors suggest a useful and important approach to the patients with PCD. The question, which PCD patients will have benefit of this approach remains open and this could be a subject of another, possibly prospective study.
Author Response
Reviewer Comment: PCD is a rare and difficult for diagnosis condition. It comprises variety of phenotypes and multiple genetic variants. That is why every study, providing additional information on the topic is valuable. I think that the authors suggest a useful and important approach to the patients with PCD. The question, which PCD patients will have benefit of this approach remains open and this could be a subject of another, possibly prospective study.
# We would like to thank the reviewer for the kind comments.

Reviewer 2 Report
I read an original article regarding primary ciliary dyskinesia (PCD), which was found to be reversible in airway obstruction. The study was conducted retrospectively on 115 patients with PCD, the diagnosis being based on clinical diagnosis in Israeli patients. For each case, the presence or absence of consanguineous marriage, height weight and BMI were examined. The study was then performed on the basis of a 10% or greater improvement in FEV1% for reversibility of airway obstruction. Many of the patients studied were already routinely treated with inhaled steroid therapy and bronchodilators.
Below are the reviewers' comments on this article. Please read the comments and respond to the authors' opinions and thoughts.
Major1.
JCM is a general medical journal, so please elaborate on the overall PCD. Genes should also be mentioned. About Kartagener's syndrome, about visceral retroversion, about hydrocephalus and odor dysesthesia.
Major2.
As a final conclusion, please state more clearly whether bronchodilators were effective, ineffective, or unknown. Also, please clearly state whether inhaled steroids were effective.
Major3
In Graph 2, please test each bar graph to see if there is a significant difference. Or add standard deviations to the bars.
Minor1
P1 L34 PCD is a recessive genetic disorder and is more frequent in consanguineous marriages. It may be more common in the Middle East and less common in Asia.
Minor2
The diagnosis of PCD is based on the electron microscopic or genetic diagnosis of a defect in the dynein arm, depending on the guidelines. Please mention this in the introduction or in P2, L52-56.
Minor3.
Is it necessary to mention smoking history in Table 2?
Minor 4
Table 2 mentions neonatal RDS, but neonatal RDS is a surfactant deficiency disease of low birth weight infants. It is a different condition from transient respiratory failure in the neonatal period, which occurs in PCD. We request a correction in this regard.
Minor 5
Macrolide antibiotics and ST combination drugs need to be mentioned as treatment. The need for early pneumococcal and type B Haemophilus influenzae immunization should also be mentioned.
Minor 6
The words "Axis Title" should be deleted from Graph 2.
Minor 7
There are no recommendations for treatment with inhaled corticosteroids, N-acetylcysteine, or high-dose gammaglobulin. What is your opinion on this point?
Best regards,
Dr. Reviewer
Round 2
Reviewer 2 Report
I have seen the revised manuscript by the authors. We received appropriate comments on 10 questions that were a bit daunting for the reviewers. The manuscript has been revised to better reflect the comments. Thank you also for the modification of the graphs.
Best regards,
Dr. Reviewer